# Efficacy of the EDP-M Scheme Plus Adjunctive Surgery in the Management of Patients with Advanced Adrenocortical Carcinoma: The Brescia Experience

**DOI:** 10.3390/cancers12040941

**Published:** 2020-04-10

**Authors:** Marta Laganà, Salvatore Grisanti, Deborah Cosentini, Vittorio Domenico Ferrari, Barbara Lazzari, Roberta Ambrosini, Chiara Sardini, Alberto Dalla Volta, Carlotta Palumbo, Pietro Luigi Poliani, Massimo Terzolo, Sandra Sigala, Guido Alberto Massimo Tiberio, Alfredo Berruti

**Affiliations:** 1Medical Oncology Unit, Department of Medical and Surgical Specialties, Radiological Sciences and Public Health, University of Brescia at ASST Spedali Civili, 25123 Brescia, Italy; m.lagana001@unibs.it (M.L.); salvatore.grisanti@unibs.it (S.G.); d.cosentini@unibs.it (D.C.); vittorio.ferrari@asst-spedalicivili.it (V.D.F.); milaro66@libero.it (B.L.); c.sardini@unibs.it (C.S.); alberto.dallavolta@gmail.com (A.D.V.); 2Radiology Unit, ASST Spedali Civili, 25123 Brescia, Italy; roberta.ambrosini@asst-spedalicivili.it; 3Urology Unit, Department of Medical and Surgical Specialties, Radiological Science and Public Health, University of Brescia at ASST Spedali Civili of Brescia, 25123 Brescia, Italy; c.palumbo003@unibs.it; 4Pathology Unit, Department of Molecular and Translational Medicine, University of Brescia at ASST Spedali Civili di Brescia, 25123 Brescia, Italy; luigi.poliani@unibs.it; 5Internal Medicine, Department of Clinical and Biological Sciences, San Luigi Hospital, University of Turin, Orbassano, 10043 Turin, Italy; massimo.terzolo@unito.it; 6Section of Pharmacology, Department of Molecular and Translational Medicine, University of Brescia, 25123 Brescia, Italy; sandra.sigala@unibs.it; 7Surgical Unit, Department of Medical and Surgical Specialties, Radiological Sciences, and Public Health, University of Brescia, ASST Spedali Civili di Brescia, 25123 Brescia, Italy; guido.tiberio@unibs.it

**Keywords:** adrenocortical tumor, treatment, EDP, mitotane

## Abstract

Etoposide, doxorubicin and cisplatin plus oral mitotane (EDP-M) comprise the reference regimen in the management of patients with adrenocortical carcinoma (ACC). In this paper, we described the outcome of 58 patients with advanced/metastatic ACC consecutively treated with EDP-M in a reference center for this rare disease in Italy. In this series, EDP-M obtained a partial response in 50% of patients; median progression free survival (PFS) and overall survival were 10.1 months (95% Confidence Interval [CI 95%] 8.1–12.8) and 18.7 months (95% CI: 14.6–22.8), respectively. EDP-M was not interrupted in five patients showing disease progression after two cycles without the appearance of new lesions and mitotane levels below the therapeutic range. In two of them, the disease remained stable at further imaging evaluations and the other three obtained a partial response. Twenty-six responding patients underwent surgery of residual disease and 13 of them became disease free. Surgery identified a pathological complete response (pCR) in four patients (7%) and Ki67 expression in post-chemotherapy tumor specimens, inferior to 15% (median value), was associated with better PFS and survival. In the present study, the EDP-M regimen is confirmed to have a limited efficacy. Early disease progression does not mean treatment inefficacy. Surgery of residual disease in partially responding patients allows for the detection of pCR in few of them and this condition is predictive of long-term survival. Ki67 expression of post-chemotherapy residual disease could be an additional prognostic factor that deserves to be studied further.

## 1. Introduction

Adrenocortical carcinoma (ACC) is a rare tumor with an estimated incidence in Western countries of 0.7–2 new cases per million population per year [1].

Surgery is the mainstay of therapy and represents the only treatment modality able to offer a chance of a cure. Unfortunately, in half of patients the disease is diagnosed at an advanced stage and most patients radically resected are destined to relapse within the first 2 years [1,2]. Advanced ACC patients, not eligible for surgery, are referred to systemic antineoplastic therapies. For decades, mitotane has been the reference drug for ACC. This compound is efficacious when serum levels reach the so-called therapeutic range (14–20 mg/L) [3], which is not usually attained before 2–3 months of therapy. The delayed efficacy of mitotane provides the main rationale for its association with chemotherapy. Chemotherapy in fact exerts a cytotoxic effect in the first months of treatment when mitotane levels are not in the therapeutic range [1]. Mitotane was also found to potentiate the cytotoxicity of certain chemotherapeutic drugs by interfering with the multidrug resistant gene 1 product P glycoprotein (Pgp), which is also expressed by ACCs [4,5]. The combination of etoposide, doxorubicin and cisplatin (EDP) is the standard chemotherapy regimen to be associated with mitotane (EDP-M) [2]. An EDP-M scheme was first introduced in 1992, when it was found to be active in the management of two young ACC patients, both obtaining a partial response lasting 7 and 21 months, respectively [6]. The feasibility of this regimen was subsequently assessed on 7 consecutive patients [7], then a formal prospective phase II trial was designed and conducted in Italy. The preliminary data of this prospective trial were published in 1998 [8] and the final results were published in 2005 [9]. In the Italian trial, the EDP-M regimen led to an overall response rate of 48.6% (95% CI: 37.1–60.3), 6.9% of patients attained a clinical complete response. The median progression free survival (PFS) and overall survival (OS) of the entire cohort were 9.1 and 28.5 months, respectively [9].

These data prompted the design and the conduction of a multicenter, multinational prospective phase III trial (the FIRM-ACT trial), in which the efficacy of the EDP-M regimen was compared to that of streptozotocyn plus mitotane (SZ-M) [10]. In this trial, patients receiving EDP-M obtained a significantly better response rate and PFS than those submitted to SZ-M. The OS also favored EDP-M (just failing to attain statistical significance), despite the crossover design of the study. The median time to progression (TTP) and OS of EDP-M treated patients were 5 months and 14.8 months, respectively, and 21% of patients were alive and free from progression after 2 years. Although the results of this multicenter randomized clinical trial revealed that EDP-M regimen has a limited efficacy in the management of advanced ACC patients, it is the only recommend treatment by current guidelines [2]. Modern therapies, such as molecular target therapy [11] and immunotherapy [12,13], have failed to demonstrate a clear efficacy in the management of ACC up to now. While we are waiting for new treatment strategies, we have to use the EDP-M scheme at its highest potential. In this paper, we present the outcome of a consecutive series of patients with advanced/metastatic ACC treated with EDP-M at a tertiary referral oncology center in Italy. The primary aim was to evaluate outcome measures such as disease response, PFS and OS, secondary aims were to evaluate the prognostic role of cytoreductive surgery, performed in patients attaining disease response and stabilization, and the prognostic role of Ki67 expression in residual disease specimens after surgery. According to this experience, we also provided some recommendations on how to best use the EDP-M regimen.

## 2. Results

### 2.1. Patients Characteristics

From January 2012 to January 2020, 58 consecutive patients with stage III or IV ACC underwent the EDP-M chemotherapy regimen at the Medical Oncology Unit at Azienda Socio Sanitaria Territoriale (ASST) Spedali Civili in Brescia. Their characteristics are summarized in Table 1.

The median age at diagnosis of ACC was 47 years (range 19–72); the male/female ratio was 18/40. At baseline conditions, six patients (10%) had unresectable locally advanced disease (stage III), the remaining 52 patients (90%) had metastatic disease (stage IV). According to modified European Network for the Study of Adrenal Tumors (ENSAT) classification (mENSAT), 25 patients (43%) presented at least two organs involved (stage IVa), 15 (26%) had three of their organs involved in the disease (stage IVb), and the remaining 12 patients (21%) had more than three tumor-involved organs (stage IVc). We also defined the Grade, R status, Age and Symptoms (GRAS) score [14] in 40 evaluable patients. Four of them (7%) were scored as GRAS grade 1, 11 patients (19%) were classified as grade 2 and 25 (43%) as grade 3. Thirty-one out of the 58 evaluable patients (53%) had hormone hypersecretion, which consisted in cortisol alone in nine patients (16%), or cortisol plus other hormones (androgens, estrogens and aldosterone) in 18 patients (31%). Two patients (3.4%) presented androgen hypersecretion alone, one patient (1.7%) presented estrogen and one (1.7%) presented aldosterone hypersecretion.

The median follow-up period for patients was 34 months (range: 12–145). At the last follow-up examination, on February 15th 2020, 38 patients (66%) were dead.

### 2.2. Treatment

A total of 280 EDP cycles were administered (median 5.5, range 1–8). Twenty-nine patients (50%) completed the treatment plan (at least six cycles); among them, 22 (38%) received six cycles, three patients (5%) received seven cycles and four patients received (7%) eight cycles.

EDP-M therapy was associated with drugs able to induce a fast decrease in serum cortisol levels in 11 patients with Cushing syndrome. In particular, five of them received metyrapone and six received abiraterone, the last ones within a prospective clinical trial (ABACUS trial NCT 03145285).

EDP-treatment was interrupted before the sixth cycle in 29 patients (50%) for the following reasons: progressive disease in 14 patients (24%), eligibility for surgical resection of the residual disease in 10 patients (17%), toxic death due to hemorrhagic shock, as a consequence of massive gastric bleeding, in one patient (2%).

Moreover, four patients discontinued the chemotherapy due to severe asthenia and performance status (PS) deterioration, despite disease stabilization, at the fifth, fifth, fourth and third cycles, respectively.

The most frequent toxicities were asthenia (grade 1 or 2) in 31 patients (53%), followed by gastrointestinal side effects, such as nausea and vomiting in 23 patients (40%), diarrhea in three patients (5.2%) and constipation in four (7%). G1/G2 hematological toxicity at treatment recycle was found in 18 patients (31%), including a reduction in the absolute number of white blood cells or platelets. Both grade 3 neutropenia and grade 4 thrombocytopenia at recycle were observed in two patients (3.4%). One death was potentially related to EDP-M scheme toxicity (hemorrhagic shock caused by massive gastric bleeding).

### 2.3. Outcome Measures

Median PFS in this series was 10.1 months (CI 95%: 8.1–12.8), (Figure 1).

The median OS was 18.7 months (CI 95%: 14.6–22.8), (Figure 2).

According to Response Evaluation Criteria in Solid Tumors (RECIST) 1.1 criteria [15], no clinical complete responses were recorded. Twenty-nine patients achieved a clinical partial response (50%), 15 (26%) a stable disease, while disease progression was observed in 14 patients (24%). Early disease progression (after one to two cycles) was observed in five patients. In particular, three of them had a deterioration of PS whereas the other two had an increase in the number of metastasis. In five additional patients, however, whose disease progression at the first radiological evaluation after two cycles was characterized by an increase in the size of metastatic lesions already present at baseline (without the appearance of new lesions), the treatment was not discontinued. Mitotane serum level was below the therapeutic range in all of them. After two additional cycles, the disease remained stable in two patients, while a partial response was observed in the remaining three. Mitotane attained the therapeutic range after three months from treatment start in two patients and after four months in the remaining three. Interestingly, one of these patients with mediastinal disease was addressed to surgery and a pathological complete response was observed at the histology of the residual disease (Figure 3).

Twenty-six patients with disease response or stabilization underwent a surgical resection of the residual disease. Twenty-five were submitted to abdominal surgery: laparotomy was adopted in 24 of them and laparoscopy in the remaining one. One patient underwent complete resection of a mediastinal metastasis. Abdominal surgery was followed by hyperthermic intraperitoneal chemotherapy treatment (HIPEC) in 14 patients. A disease-free status was attained in 13 patients that were operated on (50%)—12 after just one surgery and one after abdominal surgery followed by surgical removal of a mediastinal lesion performed 3 months later. On the contrary, a residual disease ≤10% persisted in 13 patients (50%) due to incomplete resection (R1 disease) in five of them or to the concomitant presence of small lung metastases in the remaining eight.

A post-surgery histological evaluation of the residual disease revealed a pathologically complete response (pCR) in four patients (7%). All these patients continued mitotane therapy after surgery and none of them had disease recurrence at the last follow-up examination after 30+, 27+, 10+, and 8+ months, respectively.

### 2.4. Prognostic Role of Debulking Surgery after EDP-M and Ki67 Expression in the Residual Disease

Patients who had surgery after a disease response to EDP-M obtained a better PFS than their counterparts: median 13.1 months (CI 95%: 10.3–15.9) versus 7.4 months (CI 95%: 5.5–9.3, *p* = 0.053), (Figure 4).

Similarly, the median OS was longer in surgically resected patients than those who were not: 29.8 months (95% CI:16.5–43.1) versus 10.8 months (95% CI: 7.9–13.7), respectively (*p* = 0.000), (Figure 5).

The median Ki67 expression in post-chemotherapy histology was 15 (range 1–70). Patients with Ki67 expression below the median obtained a longer PFS (15.3 months, CI 95%: 13.3–17.3) than those with residual Ki67 above or equal to the median value (11.3 months, CI 95%: 8.9–13.6; *p* = 0.025) (Figure 6). The corresponding OS medians were 21.4 months (CI 95%: 18.9–23.9) and not reached, respectively (*p* = 0.073) (Figure 7). In the same patient subset, we also evaluated the Ki67 expression at baseline. The median value was 20 (range 10–50). In 10 patients (58%), there was a reduction in Ki67 values after chemotherapy, in 17 (35%) an increase, while in one (6%), the marker expression was unchanged. In this responding patient subset, baseline Ki67 above the median was associated with a longer PFS than Ki67 expression below or equal the median value (15.1 months [95% CI: 14.5–15.7] versus 12.3 months [95% CI: 1.1–23.5], respectively, *p* = 0.034), while no difference in OS was observed (*p* = 0.788).

## 3. Discussion

In this series of advanced/metastatic ACC patients, uniformly treated in a reference oncology center for this rare disease in Italy, the EDP-M regimen was confirmed to be active. Both the overall response rate (ORR) according to RECIST criteria (50%) and the PFS (median 10 months) observed in this study were higher than the ones reported with EDP-M in the FIRM-ACT trial and were similar to those observed in the Italian phase 2 trial [9,10]. Moreover, the OS of the present ACC series was slightly longer than that of the FIRM-ACT trial, but inferior to the one observed in the published phase II study. These differences are not surprising, the efficacy of systemic antineoplastic therapies in oncology literature is often heterogeneous depending on patient selection and it appears higher in monocentric series and in phase II trials than in large multicentric clinical trials. In addition, in this series, the EDP-M was associated with side effects that were manageable in the majority of cases, although four non-progressing patients interrupted the treatment before the planned six cycles due to intense asthenia and PS deterioration, and one toxic death was seen. The effect of EDP-M in controlling hypercortisolism was not reported, since drugs able to induce a rapid decrease in cortisol levels, such as metyrapone and abiraterone [16,17,18], were associated to the scheme in 11 patients with Cushing syndrome.

The present series confirms that the EDP-M scheme has a limited efficacy but still remains the most effective therapy we actually have in the management of advanced ACC. On the basis of this single center experience, we can provide some recommendations on how to best use the EDP-M regimen in routine clinical practice.

In our series, five patients, who experienced an increase in the size of metastatic lesions already present at baseline did not interrupt the treatment plan. None of them showed disease progression at a subsequent CT scan; on the contrary, three obtained a partial disease remission and a complete pathological response was observed in one of them at subsequent histological evaluation after surgery. It is possible that this favorable outcome would not have been achieved if patients had suspended the treatment plan and had received a second-line therapy. In the clinical routine practice, an antineoplastic therapy is usually interrupted in the case of tumor progression according to the RECIST criteria. Our data suggest that this could be not the case in ACC patients receiving EDP-M, because disease control is subordinate not only to the cytotoxic effect of chemotherapy, but also to the achievement of mitotane therapeutic range, which usually occurs after 2–3 months from the beginning of the therapy. Therefore, early progression of ACC to EDP-M does not always mean treatment inefficacy; as a matter of fact, three patients with early disease progression obtained a partial response subsequently, in conjunction with the achievement of mitotane concentrations within therapeutic levels. Our data are consistent with a recently published case report [19].

The present paper also underlines the importance of a multidisciplinary approach in the management of ACC. All patients in our series without disease progression after 4–6 months were discussed by our multidisciplinary team and were pinpointed for surgery if a complete surgical removal of the disease was deemed feasible or the residual disease was estimated to be ≤10%.

As expected, patients who underwent surgery had a better outcome in terms of PFS and OS than those who did not. Of course, this does not mean that surgery in this setting is per se efficacious, since patients earmarked for surgery were selected among those already destined to have a good prognosis. However, post-chemotherapy surgery provided the advantage of reassessing the tumor after therapy. In our study, four patients (7%) attained a pathological complete response and none of them had recurred at the last follow-up. A pCR after EDP-M has already been described by our group in a single patient [20], who obtained a long-term survival. The finding of a 7% pCR rate in this series suggests that EDP-M could be very efficacious in a minority of ACC patients. pCR after neoadjuvant chemotherapy in breast cancer patients is associated to a better outcome and may be a potential surrogate of treatment efficacy [21]. Along these lines, the detection of pCR could identify a small subgroup of ACC patients who are destined to obtain a long-term benefit from the therapy.

Another advantage of post-chemotherapy surgery is the reassessment of the proliferation activity.

Ki67 expression is a major prognostic parameter in ACC [22]. In this series, we evaluated, for the first time, its prognostic role in residual tumors after EDP-M and observed that this marker, categorized by the median value, identified two patient groups with divergent PFS and OS. In early breast cancer patients, Ki67 immunostaining after neoadjuvant chemotherapy was found to have a stronger prognostic significance than its expression at baseline conditions [23]. In addition, Ki67 is a good intermediate marker to predict the efficacy of neoadjuvant hormone therapy [24,25]. Therefore, Ki67 at residual histology after EDP in ACC patients could have the same significance as in breast cancer and this issue deserves to be further explored. Noteworthily, in the patient subset who had surgery after EDP-M, who were carefully selected, we also evaluated the prognostic role of Ki67 assessed at baseline and an opposite result was observed, i.e., patients with higher Ki67 had a better prognosis. It should be noted that these patients were selected among those attaining a disease response after EDP-M and it is well known that chemotherapy efficacy is directly correlated with tumor proliferative activity. This is a possible explanation for the discrepant prognostic role of Ki67 observed before and after EDP-M. It should be noted, in any case, that baseline Ki67 in the majority of these patients was assessed in specimens obtained by a Tru-Cut biopsy and this has limited the accuracy of the assessment.

## 4. Patients and Methods

This is a monocentric, retrospective study. Consecutive ACC patients treated with the EDP-M scheme at ASST Spedali Civili of Brescia were included.

All patients met the conventional eligibility criteria adopted for chemotherapy in adult ACC patients: age >18 years; Eastern Cooperative Oncology Group (ECOG) PS0–2; pathological diagnosis of ACC; locally advanced or metastatic disease not suitable for surgery; at least one unidimensional (RECIST criteria) measurable lesion; adequate bone marrow reserve (neutrophils ≥1500/mm3 and platelets ≥100,000/mm3, hemoglobin ≥9.0 g/dl); total bilirubin ≤1.5 times the upper limit of normal; serum creatinine ≤1.5 the upper limit of normal; effective contraception in premenopausal female and male patients; written informed consent. The patient could have had a history of prior malignancy radically resected with no evidence of disease for at least 3 years.

Exclusion criteria were the contraindications to chemotherapy, such as active clinically serious infections (greater than grade 2 National Cancer Institute Common Toxicity Criteria (NCI-CTC) version 3.0); decompensated heart failure (ejection fraction ≥45%); myocardial infarction or revascularization procedure during the previous 6 months; unstable angina pectoris; uncontrolled cardiac arrhythmia; hypertension not controlled by medications; pregnancy or breast-feeding conditions.

EDP-M was administered with the following scheme: etoposide 100 mg/m^2^ (day 2–3–4), doxorubicin 20 mg/m^2^ (day 1), cisplatin 40 mg/m^2^ (day 3–4) [9]. Intravenous chemotherapy administration was performed every 4 weeks, for a maximum of 6-8 cycles. Treatment was delayed in the case of a neutrophil count <1500/mm^3^, platelet count <100,000/mm^3^, or grade >2 extra- hematological toxicity on treatment recycle. Moreover, a dose reduction was planned in the case of a neutrophil count <500/mm^3^ and/or platelet count <30,000/mm^3^ experienced at recycle. All patients received a prophylactic treatment with long-acting granulocyte colony-stimulating factor (G-CSF) after every chemotherapy cycle (administered 48 h after the end of the cycle, by subcutaneous injection).

From 2016 onwards, drugs able to induce a rapid reduction of serum cortisol levels such as metyrapone [16] or abiraterone [17,18] were associated to the EDP-M in the first month of therapy.

Oral mitotane was administered concomitantly with chemotherapy at a starting dose of 1500 mg daily, with further progressive dose increments up to the maximum tolerated dose. Serum mitotane was monitored every 4 weeks and when the patients attained the therapeutic range, the mitotane dose was tapered to maintain serum concentration within 14 and 20 mg/L.

CT scan and/or MRI were performed at baseline and every two cycles. Radiological response was assessed according to RECIST criteria [26]. Early PD after the first two cycles was not considered a strict criteria of treatment interruption. At each imaging reassessment, the patient was evaluated by a multidisciplinary team composed of medical oncologists, internal medicine physicians, surgeons and radiologists to discuss the further steps. A surgical approach with radical intent was decided if the patients attained a partial response or a stable disease after EDP-M and their metastatic disease appeared suitable for surgery. Cytoreductive surgery was also considered in patients with residual disease estimated to be ≤10%. For patients who underwent surgery, the Ki67% value at residual disease was assessed.

All included patients had given their written informed consent to the diagnostic and treatment procedures.

The following demographic, clinical, pathological and treatment data were collected: sex, age, medical history, physical examination, performance status, routine laboratory tests, endocrine work-up, chest and abdominal CT scan.

The stage of disease was assessed using the modified ENSAT (mENSAT) classification: stage III (invasion of tissues/organs or the vena renalis/cava) and stage IVa, IVb, IVc (2, 3 or>3 metastatic organs, including N, respectively) [14].

We categorized the patients according to the GRAS score, considering grading, R status, age and symptoms, defined as tumor- or hormone-related symptoms at diagnosis [14]. GRAS score was defined as GRAS 1 (favorable): ≤1 risk factor in GRAS parameters; GRAS 2 (unfavorable): 2 risk factors in GRAS parameters; GRAS 3 (pejorative): ≥3 risk factors in GRAS parameters.

PFS was defined as the time elapsed from the beginning of the EDP-M treatment until disease progression or death. Non-progressing patients still alive were censored at the last follow-up examination.

OS was defined as the time interval between the date of EDP-M treatment start and the date of death from any cause or the last known date when the patient was alive.

The retrospective study was approved by the Ethical Review Board of ASST-Spedali Civili in Brescia (protocol no 3229; version 1.1, 1st December 2019).

Descriptive statistics were used to analyze the patients’ clinical characteristics. Survival curves were calculated by the Kaplan–Meier method and differences were compared by the log rank test.

Statistical significance was set at *p* < 0.05. SPSS v23.0 software was used for the statistical analyses (SPSS Inc., Chicago, IL, USA).

## 5. Conclusions

EDP-M is confirmed as having a limited efficacy in the management of advanced/metastatic ACC patients, although a small proportion of patients can obtain a long-term clinical benefit. Early progression to EDP-M does not always mean treatment inefficacy and treatment discontinuation would not be advisable in progressing patients without the appearance of new lesions, in whom serum mitotane levels have not reached therapeutic concentrations. Surgery of residual disease in responding patients allows for the detection of pCR in few of them and this condition is predictive of long-term clinical outcomes. The reassessment of proliferation activity by Ki67 immunostaining in post-chemotherapy surgical specimens could be an additional prognostic parameter that deserves to be studied further.

## Figures and Tables

**Figure 1 cancers-12-00941-f001:**
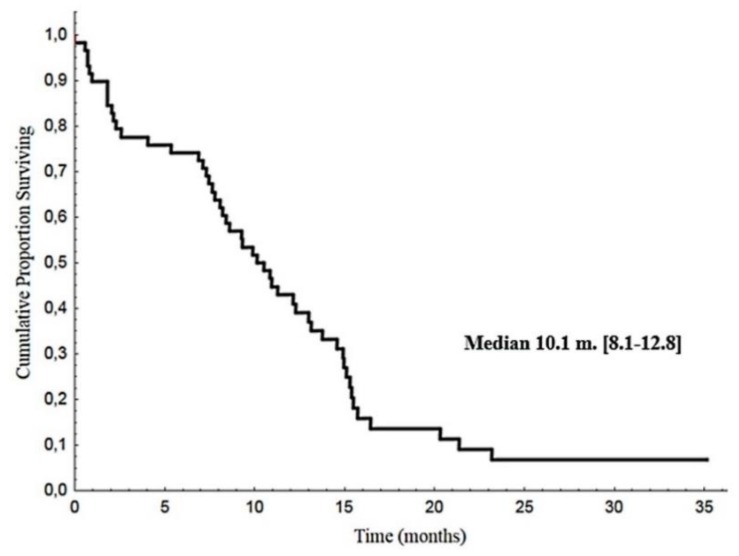
Progression-free survival.

**Figure 2 cancers-12-00941-f002:**
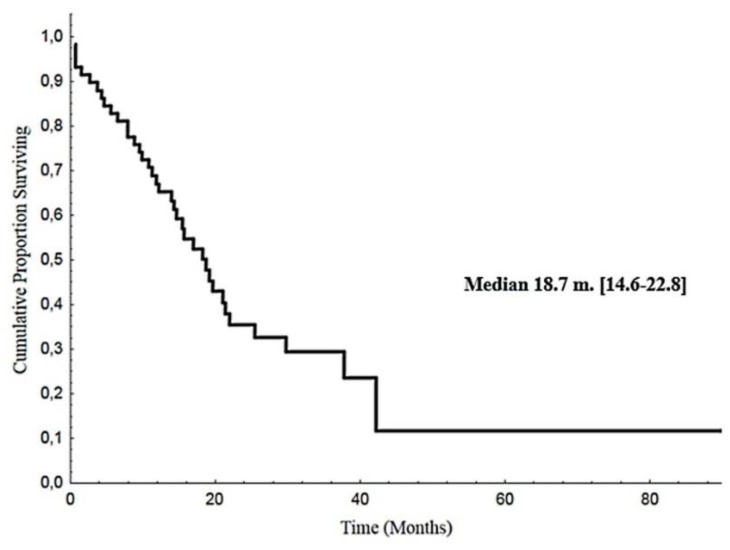
Overall survival.

**Figure 3 cancers-12-00941-f003:**
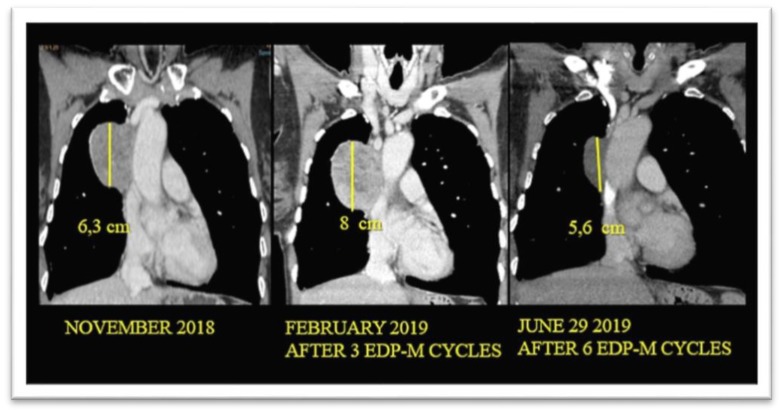
Thorax CT scan series of a patient attaining a pathological complete response after six cycles of etoposide, doxorubicin and cisplatin plus oral mitotane (EDP-M).

**Figure 4 cancers-12-00941-f004:**
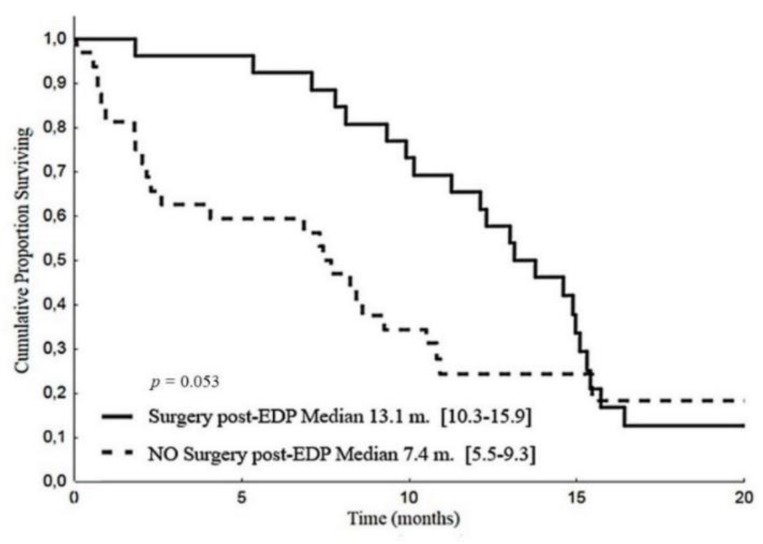
Progression-free survival according to post-EDP surgery.

**Figure 5 cancers-12-00941-f005:**
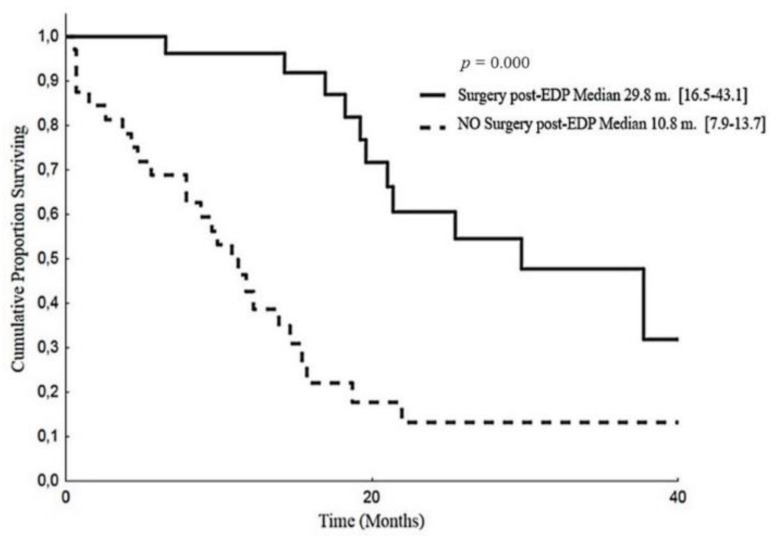
Overall survival according to post-EDP surgery.

**Figure 6 cancers-12-00941-f006:**
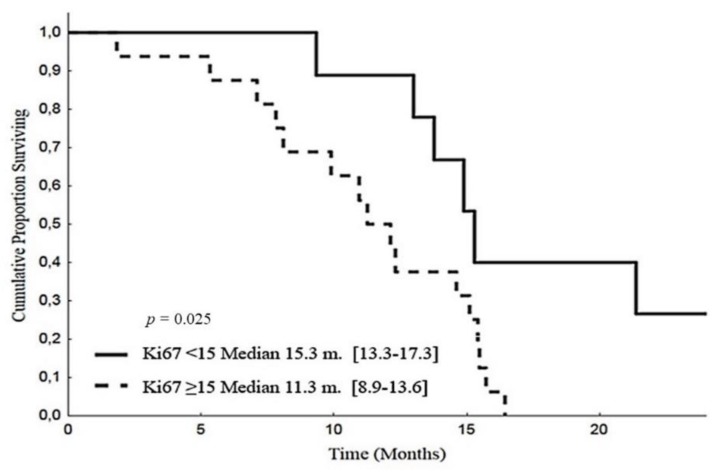
Progression-free survival according to Ki67 after post-EDP surgery.

**Figure 7 cancers-12-00941-f007:**
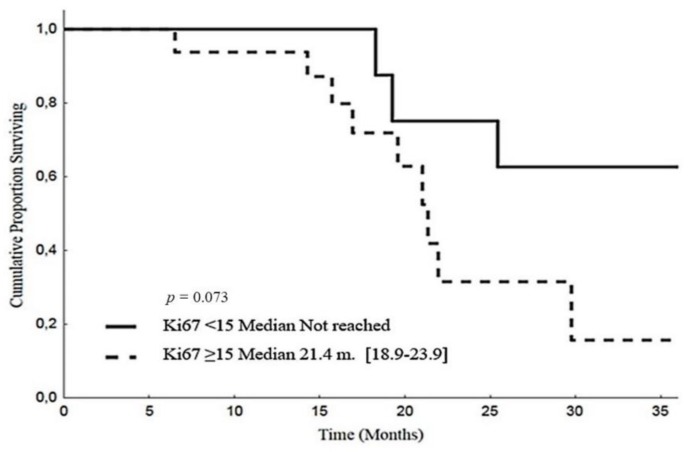
Overall survival according to Ki67 assessed after post-EDP surgery.

**Table 1 cancers-12-00941-t001:** Patients’ characteristics.

Patients’ Characteristics	Number	Percentage
**No of patients**	58	(100%)
**Median Age (range)**	47	[range 19–72]
**Male/Female ratio**	18/40	(69%)
**mENSAT stage at Diagnosis**		
III	6	(10%)
IVa	25	(43%)
IVb	15	(26%)
IVc	12	(21%)
**Hormone Hypersecretion at Diagnosis**		
Cortisol alone	9	(16%)
Cortisol plus other hormones	18	(31%)
Androgens alone	2	(3.4%)
Estrogens alone	1	(1.7%)
Aldosterone alone	1	(1.7%)
**Previous treatments: Surgery**	35	(60.3%)
**Adjuvant Mitotane**	26	(44.8%)
**Mitotane for advanced disease**	13	(22.4%)
**GRAS score* (40 available patients)**		
Favorable	4	(7%)
Unfavorable	11	(19%)
Pejorative	25	(43%)

* Grade, R status, Age and Symptoms (GRAS).

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
