# Peer review of "Efficacy of the EDP-M Scheme Plus Adjunctive Surgery in the Management of Patients with Advanced Adrenocortical Carcinoma: The Brescia Experience"

_cancers, 2020, doi:10.3390/cancers12040941_

Round 1

Reviewer 1 Report

Dear author,

I have read the article entitled, 'Efficacy of the EDP-M scheme plus adjunctive surgery in the management of patients with advanced adrenocortical carcinoma: the Brescia experience' with high interest. The article is of high priority of importance and demonstrated novel results of patient data with EDP-M scheme plus adjunctive surgery in the management of patients with advanced adrenocortical carcinoma. Hence, it is recommended to accept this manuscript after minor revisions.

Minor suggestions: There are many typographical errors throughout the maunscript, too many paragraph division is sometime distracting. Minor English and grammatical corrections also required.

Author Response

Thank you very much for the positive evaluation, we carefully read the manuscipt and correct the typographical mistakes.

Reviewer 2 Report

In this manuscript, Lagana et al. report a series of 58 patients with advanced/metastatic adrenocortical carcinoma (ACC) treated with EDP-M at a single center (Brescia, Italy). The authors assess outcome measures of response to treatment, such as progression-free survival (PFS) and overall survival (OS). Additionally, the authors evaluated the outcomes of patients who underwent cytoreductive surgery after presenting an initial response to EDP-M. Given that ACC is a rare disease and multiple gaps in therapeutic strategies for advanced cases persist, this manuscript is of interest to medical teams who treat ACC patients worldwide. The authors are leading experts in the field with a vast clinical experience in systemic polychemotherapy for advanced ACC. Although the concept is not new, since previous reports on EDP-M for ACC are available, including phase 2 and 3 trials, this report brings unique nuances that are worthy of sharing with the scientific community and are important for managing ACC patients on this drug regimen.

I have two minor comments (and questions):

One of the most interesting observations in this manuscript is that five patients who initially (after the initial two cycles) presented with an increase in metastatic masses already present at baseline obtained partial responses/disease stabilization upon the continuation of EDP-M. The authors attribute this observation to suboptimal therapeutic levels of mitotane, which often take several weeks to reach the target range. However, it is not clear whether mitotane ever reached therapeutic levels upon the continuation of EDP-M. Furthermore, would a different decision regarding the continuation of EDP-M be taken if therapeutic levels of mitotane were already present?
Another interesting observation of this manuscript if the potential benefit of cytoreductive surgery in patients who responded to EDP-M. The authors identified that a complete pathologic response and a low Ki-67 score (<15/hpf) in the residual tumor are favorable prognostic markers. It would be interesting to assess whether the pre-treatment Ki-67 score predicts response to EDP-M and how it compares to the residual tumor Ki-67 score. If this data is available, please provide it.

Author Response

Thank you very much for the positive evaluation.

Regarding your questions:

a) We better specified in the text that in all of this patients mitotane levels reached the therapeutic range after three and four months (see lines 170-171).

b) We didn’t change our chemotherapy approach according to the achievement of mitotane therapeutic range or not. We early stopped the treatment only in case of disease progression, characterized by the appearance of new lesions or increasing tumor size associated to a performance status deterioration. (see lines 163-168).

c) As requested by the reviwer we evaluated the prognostic role of Ki67 value at baseline in patients in which the marker was assessed after post EDP-M surgery.

The finding showed an opposite trend: high Ki67 level = better prognosis in term of PFS but not OS.

This finding could be due to the fact that we selected a patient population obtaining a disease response to EDP-M and it is well known that chemotherapy is more efficacious in higher proliferating tumor.

This was specified in the text and a comment was added in the discussion. (see lines 211-216; 279-287)

Reviewer 3 Report

The manuscript is prepared carelessly, its have many topological mistake.

Author Response

We apologize for the grammar and typo errors, Thank you for the remark.

We carefully checked and corrected them in the text.

Reviewer 4 Report

Well written, clearly presented. Although retrospective, studies such as this are needed in the case of extremely rare diseases like ACC. 

Author Response

We are grateful to the reviewer for the positive considerations.

Round 2

Reviewer 3 Report

I recommend the manuscript to publication in Cancer